Effect of basal luteinizing hormone/follicle-stimulating hormone ratio on clinical outcome of In Vitro fertilization in patients with polycystic ovarian syndrome: a retrospective cohort study

Shan Dan
Zhao Junzhao
Lu Xiaosheng
Zhang Huina
Lu Jieqiang jieqianglu@126.com
Shen Qi wzlunwen@163.com
Department of Gynecology and Obstetrics, the Second Affiliated Hospital and Yuying Children’s Hospital of Wenzhou Medical University , Wenzhou, Zhejiang , China
Barbosa Neto Octavio
Electronic publication date: 2024 Nov 26
Publication date: 2024
Volume: 12
Electronic Location ID: e18635
Received 2024 Aug 23; Accepted 2024 Nov 12
Copyright: © 2024 Shan et al.
Copyright year: 2024
Copyright holder: Shan et al.
License: This is an open access article distributed under the terms of the Creative Commons Attribution License, which permits unrestricted use, distribution, reproduction and adaptation in any medium and for any purpose provided that it is properly attributed. For attribution, the original author(s), title, publication source (PeerJ) and either DOI or URL of the article must be cited.
License URL: https://creativecommons.org/licenses/by/4.0/

Keywords: Gonadotropin-releasing hormone agonist protocol, IVF, LH/FSH, Mild stimulation protocol, Live birth rate, PCOS

Funding: Medical Doctor Scientific Research This work was supported by the Medical Doctor Scientific Research Start-up Fund. The funders had no role in study design, data collection and analysis, decision to publish, or preparation of the manuscript.

==============================
Background

The basal luteinizing hormone (LH) and the prior LH to follicle-stimulating hormone (FSH) ratio (LH/FSH) in polycystic ovarian syndrome (PCOS) are generally higher than those in non-PCOS patients and the general population. The potential negative effects of elevated LH on human reproductive function are highly controversial, as are the effects of down-regulation of LH on reproductive function. The purpose of this study was to evaluate the effect of the basal LH/FSH ratio on the live birth rate of PCOS patients undergoing in vitro fertilization (IVF) cycles.

Methods

A retrospective analysis was conducted on 698 patients with polycystic ovary syndrome undergoing IVF treatments with a mild stimulation protocol (n = 95) and a gonadotropin-releasing hormone (GnRH) agonist protocol (n = 603). The basal LH/FSH ratio of 2 was used as the cut-off value for further subgroup analysis. The demographic properties, controlled ovarian hyperstimulation (COH) processes, and clinical pregnancy outcomes were compared between groups under each ovulation stimulation protocol.

Results

The live birth rate for patients with a LH/FSH ratio ≥ 2 group (56.38%, n = 149) was not statistically different from that of the ones with a ratio < 2 (53.74%, n = 454) in the GnRH agonist protocol (P = 0.576). Correspondingly, the live birth rate for the LH/FSH ratio ≥ 2 group (43.48%, n = 23) also showed no statistical difference from the ratio < 2 group (48.61%, n = 72) in the mild stimulation protocol (P = 0.668). Additionally, LH/FSH ratios had no significant effect on the live birth rate after adjusting for confounders both in the GnRH agonist protocol (adjusted OR: 1.111; 95% CI [0.467–2.642], P = 0.812) and in the mild stimulation protocol (adjusted OR: 4.057; 95% CI [0.431–38.195], P = 0.221). Furthermore, there was no significant difference in the live birth rate between different ovulation stimulation protocols in PCOS patients with the LH/FSH ratio ≥ 2.

Conclusions

The live birth rate in IVF outcomes was not affected by an elevated basal LH/FSH ratio in patients with polycystic ovary syndrome. The choice of the GnRH agonist protocol or mild stimulation protocol for ovulation stimulation does not affect the final clinical outcomes either for PCOS patients with a basal LH/FSH ratio ≥ 2.

Introduction

Polycystic ovarian syndrome (PCOS) exhibits endocrine and metabolic disorders among women of reproductive age, such as anovulation, infertility, hypertrichosis, hyperandrogenemia, insulin resistance, glucose metabolism, and lipid metabolism disorders. Genetic factors and environmental factors working together may lead to the development of PCOS (Dapas & Dunaif, 2022; Siddiqui et al., 2022). The Rotterdam Criteria is usually used for the diagnosis of polycystic ovary syndrome. However, the use of the luteinizing hormone (LH) and the prior LH to follicle-stimulating hormone (FSH) ratio (LH/FSH) during ovarian stimulation remains controversial in PCOS. About 74.7% of Chinese patients have abnormal LH levels and LH/FSH ratios, compared with about 80% or more in Japan and 39.8% in the U.S (Chen et al., 2008). These data suggest that Southeast Asian and Chinese patients are significantly different from Western patients in terms of abnormalities in LH levels and LH/FSH ratios. The Rotterdam Criteria does not include the LH/FSH ratio as one of the diagnostic criteria for PCOS, but the Reproductive Endocrine Committee of the Japanese Obstetrics and Gynecology Association regards LH/FSH ≥ 1 as an important indicator for the diagnosis of PCOS (Hendriks et al., 2008). Some scholars believe that the Japanese PCOS standard is more suitable for Asian populations. Some scholars have proposed that LH/FSH has important reference value for the diagnosis of suspected polycystic ovary syndrome (Le et al., 2019). It is generally believed that the LH/FSH ratio in PCOS is greater than 2–3, which causes endocrine abnormalities and makes the hypothalamic-pituitary-ovarian axis adjustment function abnormal. Studies have shown that any increase in this ratio depends on factors such as race, age, and body mass index (BMI) (Bozdag et al., 2016; Saadia, 2020). Normal follicular development depends on a certain level of LH. When LH is below the LH threshold, the oocyte does not completely mature because there is insufficient androgen and estrogen synthesis and a lack of paracrine signaling between the granulosa cells and the theca cells. When LH is above the “upper limit,” the proliferation of the granulosa cells is inhibited because the follicle is atretic or prematurely luteinized, and the embryo quality and pregnancy outcomes are affected (Turathum, Gao & Chian, 2021). High basal LH levels may have negative effects on IVF/ICSI treatment outcomes (Jiang et al., 2020). Studies have shown a positive correlation between AMH and LH/FSH ratio, especially in lean PCOS patients (Pratama et al., 2024). AMH is thought to be a receptor that acts on GnRH neurons to increase the LH/FSH ratio (Dilaver et al., 2019). It has been suggested that high bLH/FSH levels may lead to abnormal granulosa cell function and inhibit ovarian follicle formation (Xie et al., 2019).

It is still unknown whether patients could really benefit from the suppression of a high basal level of LH in advance. Studies on oral contraceptive pills (OCP) pretreated PCOS patients with elevated basal LH to reduce their LH level to normal before the controlled ovulation stimulation IVF cycle or intracytoplasmic sperm injection (ICSI) cycle and compared the clinical outcomes. Some studies found that the fertilization rate, embryo implantation rate, and clinical pregnancy rate were significantly improved with oral contraceptive pills (Pan et al., 2015), while some studies have found that OCP pretreated PCOS patients did not improve the clinical pregnancy rate, and even increased the miscarriage rate (Fernandez-Prada et al., 2022; Kalem, Kalem & Gurgan, 2017). A full dose of GnRH agonist is a kind of drug that lowers the levels of endogenous gonadotropins such as LH and FSH. The current study aimed to show the effects of the LH/FSH ratio on the final clinical outcome of IVF in polycystic ovary syndrome patients with the cut-off value of LH/FSH equal to two according to the guidelines for the diagnosis and treatment of polycystic ovary syndrome in China in 2018. Do different ovulation stimulation protocols affect the final clinical outcome for PCOS patients with a high basal LH/FSH ratio?

Materials and Methods

A retrospective analysis was conducted on 698 patients with polycystic ovary syndrome undergoing IVF in the Reproductive Center of the Second Affiliated Hospital of Wenzhou Medical University from July 2018 to June 2022. The inclusion criteria included: All patients satisfied the Rotterdam diagnostic criteria (Rotterdam ESHRE/ASRM-Sponsored PCOS Consensus Workshop Group, 2004), were less than 40 years old, and had tubal factor infertility in addition to polycystic ovary syndrome. The exclusion criteria included: hypertension, diabetes, endometriosis, adenomyosis, obvious endometrial polyps, uterine malformations, intrauterine adhesions, hyperprolactinemia, and male factors. A total of 95 women received mild stimulation protocols, while 603 women received GnRH-agonist protocols. The basal LH/FSH ratio equal to 2 (LH/FSH = 2) was considered as the cut-off point in the study (Li et al., 2010; Orvieto et al., 2012). All patients included in the study were divided into four groups with a ratio of LH/FSH equal to two and two protocols for ovulation stimulation: 72 cycles with LH/FSH < 2 and 23 cycles with LH/FSH ≥ 2 in the mild stimulation protocol; 454 cycles with LH/FSH < 2 and 149 cycles with LH/FSH ≥ 2 in the agonist protocol (Fig. 1). In the GnRH-agonist protocol, patients were injected with GnRH agonist (Diphereline, Ipsen, France) on day 2–4 of menstruation, and after 30–38 days when the pituitary gland reached the downregulation criteria, daily rFSH injections (Recombinant Human Follitropin, Merck, Germany) were started until and including the day of β-hCG administration. In the mild stimulation protocol, patients started daily rFSH injections (Recombinant Human Follitropin, Merck, Germany) on day 3 of menstruation until and including the day of β-hCG administration. When three or more follicles reached a mean diameter of ≥ 17 mm, 4,000–10,000 IU of β-hCG (Chorionic Gonadotropin, Lizhu, China) was administered intramuscularly. Luteal phase support was started on the day of oocyte retrieval. IVF procedures were performed as described previously. The criteria for embryo quality assessment were based on the criterion on embryo assessment (ESHRE Guideline Group on Good Practice in IVF Labs et al., 2016). Embryo transfer was performed at the blastocyst stage. The hormonal parameters of all the patients were measured by immunochemiluminescence. We compared the demographic properties, the controlled ovarian hyperstimulation (COH) processes, the number of oocytes retrieved, the embryos transferred, the high-quality embryo rate at the cleavage stage and blastocyst, the clinical pregnancy rate, the spontaneous abortion rate, and the live birth rate by different LH/FSH ratios and by different ovulation stimulation protocols. In the study, clinical pregnancy was defined as the detection of a gestational sac and fetal heartbeat in the uterus by B-Scan ultrasound about 4–5 weeks after embryo transfer. The live birth rate was the primary endpoint. This study was approved by the Ethics Committee of the Second Affiliated Hospital and Yuying Children’s Hospital of Wenzhou Medical University, China (No. 2023-K-135-01), and the exemption from informed consent was obtained from the Ethics Committee. All methods were carried out in accordance with relevant guidelines and regulations.

Figure 1 Flow chart.

A total of 698 polycystic ovarian syndrome patients undergoing IVF procedures were studied in the Reproductive Center of the Second Affiliated Hospital of Wenzhou Medical University from July 2018 to June 2022. There were 603 cycles using the gonadotropin agonist protocol and 95 cycles using the mild stimulation protocol. The basal LH/FSH ratio of 2 was used as the cut-off value for subgroup analysis.

Statistical methods

Continuous data were expressed as the mean ± standard deviation or interquartile range (IQR) depending on the distribution of the data. The differences between groups were tested by the t-test with a normal distribution or the Mann-Whitney test with a non-normal distribution. Categorical data were represented as percentages, and differences between groups were assessed by the χ2 analysis, with Fisher’s exact test for expected frequencies less than five. Pearson correlation analysis or Spearman correlation analysis was used in bivariate correlation analysis depending on the distribution of the data. Bicategorical logistic regression was applied for the regression analysis of variables. For all the tests, changes were considered significant at different confidence levels, either P < 0.05, 0.01, or 0.001, as appropriate.

Results

There were 454 cycles that underwent the GnRH agonist protocol, including 72 cycles that underwent the mild stimulation protocol with their LH/FSH ratio < 2, while 149 cycles underwent the GnRH agonist protocol and 23 cycles underwent the mild stimulation protocol with their LH/FSH ratio ≥ 2. In the present study, it was found that there was no significant difference in age, BMI, infertility duration, basal FSH and estrogen levels, total doses of gonadotropin used during the controlled ovarian hyperstimulation process, endometrial thickness and estrogen levels on the trigger day, as well as the number of oocytes retrieved in the same ovulation protocol between the two groups with the LH/FSH ratio bounded by 2 (Table 1, P > 0.05). However, the basal testosterone level was significantly higher in the LH/FSH ratio ≥ 2 group than that in the LH/FSH ratio < 2 group in the GnRH agonist protocol (Table 1, P = 0.001), while it showed the same trend regarding the basal testosterone level in the mild stimulation protocol but did not reach statistical significance. The gonadotropin administration days were significantly prolonged in the group with an LH/FSH ratio ≥ 2 compared to the group with a LH/FSH ratio < 2 (Table 1, P = 0.035). It was shown that there was no significant difference in the high-quality embryo rate at the cleavage stage or blastocyst stage both in the LH/FSH ratio ≥ 2 group and the LH/FSH ratio < 2 group in the GnRH agonist protocol (Table 1, P = 0.828, P = 0.408). Also, no significant differences in these parameters were found in the mild stimulation protocol. No significant difference in the number of embryos transferred and the high-quality blastocyst rate was found in the different LH/FSH ratio groups in both ovulation protocols (Table 1, P = 0.968, P = 0.257). The incidence of severe ovarian hyperstimulation syndrome (OHSS) was 1.98% (LH/FSH <2) and 2.68% (LH/FSH ≥ 2) in the agonist group respectively; however, it was 1.39% (LH/FSH < 2) and 0 (LH/FSH ≥ 2) in the mild-stimulation protocol group respectively, and there was no statistically significant difference in the comparison of the groups (Tables 1 and 2).

Table 1 Baseline characteristics, ovarian stimulation, and clinical outcomes across LH/FSH ratios and ovulation stimulation protocols.

	Gonadotropin agonist protocol	P value	Mild stimulation protocol	P value	
Group	LH/FSH < 2	LH/FSH ≥ 2		LH/FSH < 2	LH/FSH ≥ 2		
Cycles	454	149		72	23		
Age (years old)	29.0 (27.0, 32.0)	28.0 (27.0, 31.0)	0.600	29.82 ± 3.78	28.83 ± 3.69	0.089	
BMI (kg/m2)	23.24 (20.31, 26.30)	22.68 (20.31, 25.30)	0.238	23.93 ± 3.91	23.44 ± 3.20	0.590	
Infertility duration (years)	3.0 (2.0, 4.0)	3.0 (2.0, 5.0)	0.416	3.0 (1.0, 5.0)	2.0 (1.0, 4.0)	0.406	
Basal sex hormone level							
FSH (IU/L)	6.28 (5.50, 7.33)	6.41 (5.32, 7.54)	0.998	5.96 ± 2.00	5.57 ± 1.79	0.410	
LH (IU/L)	6.99 (5.03, 9.35)	17.91 (14.28, 22.00)	0.001***	5.37 (3.39, 8.49)	14.83 (10.29, 18.08)	0.001 ###	
Estradiol (ng/ml)	43.00 (37.00, 52.50)	46.00 (38.00, 54.00)	0.241	40.94 ± 14.07	42.76 ± 11.18	0.631	
Testosterone (ng/ml)	0.38 (0.28, 0.52)	0.46 (0.37, 0.62)	0.001***	0.44 ± 0.20	0.52 ± 0.20	0.112	
Total gonadotropin dose (IU)	1,725.1 (1,350.0, 2,281.3)	1,800.0 (1,325.0, 2,250.0)	0.938	1,387.5 (1,125.0, 1,950.0)	1,350.0 (1,050.0, 2,175.0)	0.886	
Total gonadotropin time (days)	12.0 (10.0, 14.0)	13.0 (11.0, 15.0)	0.035*	10.0 (9.0, 12.75)	10.0 (9.0, 15.0)	0.837	
Estradiol level on HCG trigger day (ng/ml)	2,169.0 (1,329.5, 2,940.5)	2,346.1 (1,658.3, 3,155.5)	0.068	1,750.5 (577.3, 2,885.5)	2,240.0 (1,281.0 3,009.0)	0.113	
Endometrial thickness on HCG trigger day (mm)	10.9 (9.8, 12.2)	11.0 (10.0, 12.2)	0.486	10.0 (9.0, 11.9)	10.3 (9.6, 11.8)	0.311	
Number of oocytes retrieved	14.0 (10.0, 18.0)	13.0 (10.0, 18.0)	0.720	12.04 ± 7.45	13.04 ± 5.01	0.549	
Number of embryos transferred	1.0 (1.0, 2.0)	1.0 (1.0, 2.0)	0.968	2.0 (1.0, 2.0)	1.0 (1.0, 2.0)	0.257	
High-quality embryo on cleavage stage rate, %(n)	43.69% (1,909/4,369)	43.36% (611/1,409)	0.828	43.79% (247/564)	46.80% (95/203)	0.460	
High-quality blastocyst rate, %(n)	37.49% (1,638/4,369)	36.27% (511/1,409)	0.408	23.23% (131/564)	25.12% (51/203)	0.586	
Severe OHSS rate, %(n)	1.98% (9)	2.68% (4)	0.406	1.39% (1)	0 (0)	0.758	
Implantation rate, %(n)	54.85% (328/598)	58.67% (115/196)	0.350	43.64% (48/110)	56.25% (18/32)	0.208	
Spontaneous abortion rate, %(n)	14.08% (40/284)	15.15% (15/99)	0.794	14.63% (6/41)	16.67% (2/12)	0.588	
Clinical pregnancy rate, %(n)	62.56% (284/454)	66.44% (99/149)	0.392	56.94% (41/72)	52.17% (12/23)	0.688	
Livebirth rate, %(n)	53.74% (244/454)	56.38% (84/149)	0.576	48.61% (35/72)	43.48% (10/23)	0.668	
Notes:

*P < 0.05, ***P < 0.001 between different LH/FSH ratio groups with the gonadotropin agonist protocol stimulation.

###P < 0.001 between different LH/FSH ratio groups with the mild simulation protocol stimulation.

BMI, body mass index; FSH, follicle-stimulating hormone; LH, luteinizing hormone.

Table 2 Baseline characteristics, ovarian stimulation characteristics and clinical outcomes in different ovulation stimulation protocols when LH/FSH ratio ≥2.

	Gonadotropin agonist protocol	Mild stimulation protocol	P value	
Group	LH/FSH ≥ 2	LH/FSH ≥ 2		
Cycles	149	23		
Age (years old)	28.0 (27.0, 31.0)	27.0 (26.0, 30.0)	0.114	
BMI (kg/m2)	23.14 ± 3.74	23.44 ± 3.20	0.718	
Infertility duration (years)	3.0 (2.0, 5.0)	2.0 (1.0, 4.0)	0.176	
Basic sex hormone level				
FSH (IU/L)	6.43 ± 1.99	5.57 ± 1.79	0.053	
LH (IU/L)	13.43 (10.71, 16.50)	11.12 (7.72, 13.56)	0.011*	
Estradiol (ng/ml)	44.10 (36.62, 53.25)	39.0 (37.3, 46.73)	0.216	
Testosterone (ng/ml)	0.46 (0.37, 0.62)	0.50 (0.40, 0.65)	0.504	
Total gonadotropin dose (IU)	1,800.0 (1,325.0, 2,250.0)	1,350.0 (1,050.0, 2,175.0)	0.056	
Total gonadotropin time (days)	13.0 (11.0, 15.0)	10.0 (9.0 15.0)	0.030*	
Estradiol level on HCG trigger day (ng/ml)	2,346.1 (1,658.3, 3,155.5)	2,240.0 (1,281.0 3,009.0)	0.731	
Endometrial thickness on HCG trigger day (mm)	11.15 ± 2.00	10.55 ± 1.52	0.168	
Number of oocytes retrieved	13.0 (10.0, 18.0)	13.0 (9.0, 17.0)	0.560	
Number of embryos transferred	1.0 (1.0, 2.0)	1.0 (1.0, 2.0)	0.471	
High-quality embryos on cleavage stage rate, %(n)	43.36% (611/1,409)	46.80% (95/203)	0.357	
High- quality blastocyst rate, %(n)	36.27% (511/1,409)	25.12% (51/203)	0.002**	
Severe OHSS rate, %(n)	2.68% (4)	0 (0)	0.560	
Implantation rate, %(n)	58.67% (115/196)	56.25% (18/32)	0.797	
Spontaneous abortion rate, %(n)	15.15% (15/99)	16.67% (2/12)	0.581	
Clinical pregnancy rate, %(n)	66.44% (99/149)	52.17% (12/23)	0.183	
Livebirth rate, %(n)	56.38% (84/149)	43.48% (10/23)	0.248	
Notes:

*P < 0.05, **P < 0.01 between gonadotropin agonist protocol group and mild stimulation group when LH/FSH ratio ≥ 2.

BMI, body mass index; FSH, follicle-stimulating hormone; LH, luteinizing hormone.

There were no statistically significant differences in clinical outcomes, including the clinical pregnancy rate, implantation rate, spontaneous abortion rate, and live birth rate between the different LH/FSH ratio groups in both ovulation stimulation protocols (Table 1, P > 0.05). Although the live birth rates were slightly higher in the LH/FSH ratio ≥ 2 group (56.38% in the GnRH agonist protocol and 43.48% in the mild stimulation protocol) than in the LH/FSH ratio < 2 group (53.74% in the GnRH agonist protocol and 48.61% in the mild stimulation protocol), none of them reached statistical significance.

Correlation test analysis revealed that the LH/FSH ratio was negatively correlated with BMI and the number of embryos transferred, and positively correlated with basal LH, estrogen, testosterone levels, and stimulate ovulation days in GnRH agonist protocol cases. Meanwhile, it was found that the LH/FSH ratio was negatively correlated with the number of embryos transferred and positively correlated with basal LH, estrogen, and testosterone levels in mild stimulation protocol cases (Table S1). After correcting for confounding factors by bicategorical logistic regression analysis in the GnRH agonist protocol group, the OR of the basic LH/FSH ratio on the live birth rate was 1.111, 95% CI [0.467–2.642], which was not statistically significant (Table 3, P = 0.812). Meanwhile, the basic LH/FSH ratio and live birth rate were not statistically significant in the mild stimulation protocol group, with an adjusted OR of 4.057, 95% CI [0.431–38.195] (Table 3, P = 0.221).

Table 3 Adjusted live birth rates by LH/FSH ratio groups based on stimulation protocols.

	Gonadotropin agonist protocol	Mild stimulation protocol	
	OR	95% CI	P	OR	95% CI	P	
LH/FSH	1.111	[0.467–2.642]	0.812	4.057	[0.431–38.195]	0.221	
Note:

OR, odds ratio; CI, confidence interval.

For all PCOS patients with a LH/FSH ratio ≥ 2 (149 cycles in the GnRH agonist protocol and 23 cycles in the mild stimulation protocol), there was no statistical difference in the comparison of demographic properties and COH process index between the two groups. It showed that the high-quality blastocyst rate was 36.27% in the GnRH agonist protocol group, which was significantly higher than that of 25.12% in the mild stimulation protocol group (Table 2, P < 0.01). While the clinical pregnancy rate, implantation rate, spontaneous abortion rate, and live birth rate were not statistically different between these two protocols with LH/FSH ratios ≥ 2 (Table 2, P = 0.002). For example, the live birth rate was 56.38% in the GnRH agonist protocol, which was higher than 43.48% shown in the mild stimulation protocol, but did not reach statistical significance (Table 2, P = 0.248). Correlation test analysis revealed that the LH/FSH ratio was negatively correlated with infertility duration, and positively correlated with basal LH and testosterone levels for patients with an LH/FSH ratio ≥ 2 (Table S2). It was found that the live birth rate was not significantly affected by the basic LH/FSH ratio in PCOS patients with a basal LH/FSH ratio ≥ 2 using different ovulation stimulation protocols by bicategorical logistic regression analysis after correcting for confounding factors, with an adjusted OR of 0.687, 95% CI [0.373–1.268] (Table S3, P = 0.230).

Discussion

In the current study, there was no statistically significant difference in the baseline FSH value in different LH/FSH ratio groups both with patients with the agonist protocol and the mild stimulation protocol. However, both the baseline LH and baseline testosterone values were significantly elevated in the high LH/FSH ratio group, and there was a significant positive correlation. The dysfunction of the H-P-O axis regulation in PCOS patients resulted in an increase in androgen levels, an increase in peripheral estrogen production, an increase in GnRH secretion pulses. There was almost no progesterone effect in PCOS patients because of prolonged anovulation, endometrial remodeling was impaired, so progesterone resistance of the endometrium occurred (Matsuyama, Whiteside & Li, 2024; Xue et al., 2021). High-pulse GnRH leads to a high expression of LH-mRNA, but not FSH mRNA expression, resulting in increased LH secretion. High levels of LH contribute to an increase in the activity of the rate-limiting enzyme 17a-hydroxypregnenolone (P450c17a) for androgen synthesis, leading to an increased secretion of androgen levels in follicular theca cells, while a decrease in the aromatase activity of follicular granulosa cells reduces the conversion of androgens to E2 and estrone (von Wolff et al., 2017). Increased levels of other hormones such as GnRH and leptin except LH are also strongly associated with the occurrence of PCOS and together resulting in hormone imbalance (Turki & Ammar, 2024).

LH plays an important role in follicular development, ovulation, and the developmental potential of oocytes (Gershon & Dekel, 2020). The normal development of a follicle requires the concentration of LH to be maintained between the minimum and maximum thresholds in the natural menstrual cycle and ovulation-promoting cycle, which is called the LH window. Follicular development can be negatively affected by either too high or too low LH. Although oocyte quality would be adversely affected by high LH levels according to endocrinological theoretical views (Gershon & Dekel, 2020). Excessive LH levels are closely related to endocrine disruption within the follicle. Elevated LH stimulates the follicular mesenchyme and follicular theca cells to secrete excessive androgens which inhibit follicular maturation and result in impaired follicular maturation and infertility (Witchel & Plant, 2020). A retrospective study showed that the clinical pregnancy rate and live birth rate were significantly lower in the high bLH/FSH group than in the control group in the fresh embryo transfer cycle (Wang et al., 2024a). However, our results showed that high LH levels did not affect the rate of high-quality embryos at the cleavage and blastocyst stages in the same ovulation protocol. This result was consistent with studies in a PCOS population (Liu & Wang, 2023; Sun et al., 2018). However, the rate of high-quality embryos at the blastocyst stage was significantly higher in the agonist protocol group than in the mild stimulation protocol when the LH/FSH ratio ≥ 2. It may be attributed to the fact that the agonist protocol is more friendly to the euploidy of embryonic chromosomes. It was shown that compared with the GnRH agonist protocol, the GnRH anta protocol was correlated with a lower euploidy rate per embryo biopsied in preimplantation genetic testing for aneuploidy cycles (Wang et al., 2024b). It was also shown that the GnRH-anta protocol was associated with higher aneuploidy rates in early abortion tissues and blastocysts compared to the GnRH-agonist protocol (Wang et al., 2022). Another explanation could be that agonists reduce the LH level of the PCOS patients from the beginning, which adjusted the endocrine disruption and promotes the beneficial follicular growth. But the real mechanism still needs to be carried out in further research. The sustained suppression of endogenous gonadotropins by the long agonist protocol may be related to slower follicular development. Therefore, longer gonadotropin days and more total doses were used in the agonist protocol. It can be found that the estrogen level on the trigger day was relatively higher in the agonist protocol group than in the mild stimulation protocol group. When there was no significant difference in the number of oocytes retrieved, the estrogen level per follicle was much higher in the agonist protocol group than in the mild stimulation protocol group, which may be the reason for the better quality of embryos at the blastocyst stage in the agonist protocol group. The incidence of severe OHSS appeared to be higher in the agonist group than in the mild-stimulation protocol group, but there was no statistically significant difference, and it may be that the sample size was not large, and we will enlarge the sample size for further study.

In regards to clinical outcomes, it was found that the clinical pregnancy rate, live birth rate, and miscarriage rate did not suffer from the negative impact of a high baseline LH value. However, the final live birth rate was slightly higher in the high LH/FSH ratio group than in the low ratio group in the agonist protocol, but it did not reach statistical significance. Our results are in agreement with some previous studies (Bansal et al., 2016; Sun et al., 2018). Some scholars pre-decreased LH levels before ovulation by drugs such as OCP (Ozmen et al., 2014) or metformin (Abdalmageed et al., 2019; Christianson et al., 2015), but IVF outcomes were not significantly changed. In the current study, the basal LH level was not reduced in advance in the mild stimulation protocol because the pituitary was not desensitized, so the LH level was relatively high at the initiation of stimulation. While it is characterized by the continuous suppression of endogenous LH levels in the GnRH agonists protocol and controlled ovarian hyperstimulation started with down-regulated endogenous LH levels, exhibiting longer stimulation times and more gonadotropin during the process of ovulation stimulation, which effectively prevents premature luteinization of follicles, promotes synchronized follicular development, and also reduces androgen levels and improves oocyte quality. Although our results showed a higher percentage of high-quality blastocysts in the agonist protocol compared to the mild stimulation protocol, there was no statistically significant difference in the final outcomes, including the live birth rate, between the two protocols for PCOS patients. The limitation of this article is the small sample size, and we will conduct a prospective study in the future to include more patients, include more indicators such as AMH because of the strong correlation between AMH and basal LH levels, and assay the follicular fluid composition of the patients to further investigate the possible mechanisms of the problem.

Conclusions

In conclusion, it was shown that the basal LH/FSH ratio in polycystic ovary syndrome had no significant effect on the clinical outcome of IVF patients. Therefore, it is deemed unnecessary to intentionally lower LH levels to a certain level before the initiation of stimulation. The number of study cycles was relatively small, and our conclusions need to be made with caution. Since no ovarian stimulation protocol is clearly superior to another, our aim is to choose the appropriate treatment based on the characteristics of the patient with polycystic ovary syndrome, to reduce the time and cost of treatment, and to achieve pregnancy as early as possible.

Supplemental Information

Supplemental Information 1 Raw data.

Supplemental Information 2 Supplementary materials.

We want to gratefully acknowledge professor Haolin Chen for the revision of the article.

Additional Information and Declarations

Competing Interests

Author Contributions

Human Ethics

Data Availability

The authors declare that they have no competing interests.

Dan Shan conceived and designed the experiments, performed the experiments, analyzed the data, prepared figures and/or tables, authored or reviewed drafts of the article, and approved the final draft.

Junzhao Zhao conceived and designed the experiments, analyzed the data, prepared figures and/or tables, authored or reviewed drafts of the article, and approved the final draft.

Xiaosheng Lu conceived and designed the experiments, performed the experiments, analyzed the data, prepared figures and/or tables, authored or reviewed drafts of the article, and approved the final draft.

Huina Zhang conceived and designed the experiments, performed the experiments, analyzed the data, prepared figures and/or tables, authored or reviewed drafts of the article, and approved the final draft.

Jieqiang Lu conceived and designed the experiments, analyzed the data, prepared figures and/or tables, authored or reviewed drafts of the article, and approved the final draft.

Qi Shen conceived and designed the experiments, performed the experiments, analyzed the data, prepared figures and/or tables, authored or reviewed drafts of the article, and approved the final draft.

The following information was supplied relating to ethical approvals (i.e., approving body and any reference numbers):

the Ethics Committee of the Second Affiliated Hospital and Yuying Children’s Hospital of Wenzhou Medical University, China (No.2023-K-135-01).

The following information was supplied regarding data availability:

The raw data is available in the Supplementary Files 1 and 2.

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
