# Peer review of "Effect of basal luteinizing hormone/follicle-stimulating hormone ratio on clinical outcome of In Vitro fertilization in patients with polycystic ovarian syndrome: a retrospective cohort study"

_PeerJ, doi:10.7717/peerj.18635_

## Round 0.1 · original submission · Major Revisions

Dear authors,

Manuscript titled "Effect of Basal Luteinizing Hormone/Follicle-Stimulating Hormone Ratio on Clinical Outcome of In Vitro Fertilization in Patients with Polycystic Ovarian Syndrome: A Retrospective Cohort Study" that you submitted to PeerJ has been reviewed.

The reviewer(s) have suggested that some important points must be clarified and have requested substantial changes to be made in the manuscript. Therefore, I invite you to respond to the reviewer(s)' comments and revise your manuscript. The reviewer(s) comments are included at the end of this letter.

Please ensure that all review, editorial, and staff comments are addressed in a response letter and that any edits or clarifications mentioned in the letter are also inserted into the revised manuscript where appropriate.

Reviewer 1 ·

Basic reporting

This is a well-accomplished and relevant article in the field of IVF treatments in patients with PCOS. It is well structured, well formatted with a very acceptable level of English.

The association of LH elevation and the LH/FSH ratio in these patients is already well known in the literature. There are already publications that correlate these hormones with anti-Müllerian hormone (AMH), which is the marker par excellence of ovarian reserve and with growing potential in the evaluation and reproductive prognosis of women with PCOS undergoing ART treatments.

1 - In this sense, I have the following MAJOR review to propose to the authors:
The future is about the AMH; it is necessary that the introduction and discussion of this article refer to studies on AMH in this context and the relationship with the data already published and obtained by the authors on LH and the LH/FSH ratio (which were not statistically significant).

I suggest that the authors evaluate the original studies and a systematic review published by the same group, which in addition to reviewing the composition of the follicular fluid of women with PCOS in relation to all these hormones, has very interesting results on AMH as a prognostic marker and its relationship with LH and the LH/FSH ratio:

Moreira MV, Vale-Fernandes E, Albergaria IC, Alves MG, Monteiro MP. Follicular fluid composition and reproductive outcomes of women with polycystic ovary syndrome undergoing in vitro fertilization: A systematic review. Rev Endocr Metab Disord. 2023 Dec;24(6):1045-1073. doi: 10.1007/s11154-023-09819-z. Epub 2023 Jul 26. PMID: 37493841; PMCID: PMC10697886.

Experimental design

1 - It would be useful to know the relationship between the AMH of these patients and the LH assays and the LH/FSH ratio;

2 - These are women with PCOS, at risk of ovarian hyperstimulation syndrome, therefore, it is necessary to assess the ovarian reserve markers - AMH and antral follicle count - and to know the rate of cases with hyperstimulation syndrome and the strategy adopted;

3 - Why did the authors use protocols with GnRH agonists, knowing that women with PCOS, at risk of hyperstimulation syndrome, benefit from using antagonist protocols? The authors always triggered with hCG and this constituted a risk in these patients.

4 - It is necessary to characterize the drugs used in controlled stimulation of the ovaries: recombinant FSH, recombinant FSH and LH, etc., total doses, stimulation days;

5 - What was the rate of embryo cryopreservation? Can you calculate the cumulative pregnancy rate considering the number of embryo transfers by follicular puncture? Were all transfers carried out fresh? What is the value of progesterone on the day of the trigger?

Validity of the findings

Study with validity. Adequate statistical treatment.

Additional comments

Interesting and well-conducted, although retrospective study.

If it is not possible to address the previous points, their absence in the limitations of the study should be discussed.

I am convinced that the study is publishable.

·

Basic reporting

I find it great

Experimental design

Multiple arms in study leads to small sample in each arm

Validity of the findings

Couldn’t find fresh vs frozen transfers
SET vs DET results should be highlighted

Additional comments

Can plan a prospective study

·

Basic reporting

1-The language in this manuscript is strong, but it requires minor revisions. For instance, in the abstract, add (the) prior LH (41) and replace 'normal population' with 'general population' (42).
2- Please rename the following tables :
Table 1: Baseline Characteristics, Ovarian Stimulation, and Clinical Outcomes Across LH/FSH Ratios and Ovulation Stimulation Protocols
Table 3: Adjusted Live Birth Rates by LH/FSH Ratio Groups Based on Stimulation Protocols
3-Please, Give the description for figure 1

Experimental design

1- The research question in this manuscript investigates whether the LH/FSH ratio affects in vitro fertilization outcomes in patients with polycystic ovarian syndrome.
2- Choosing a retrospective cohort study was successful; however, the number of patients varied between protocols, with 95 women receiving mild stimulation protocols and 603 women receiving GnRH-agonist protocols. This leads to an important question: Which of the two methods is theoretically safer and more effective in preserving hormonal balance?
3- State the type of drugs that were used in this study

Validity of the findings

The results of the current study are both statistically and clinically valid; however, the authors must also illustrate the methods used for assessing the hormonal parameters.
Recent studies should be referenced in the discussion of the results. The following source can be cited as a recent reference:

Thualfiqar Ghalib Turki, Jumana Waleed Ammar, “Role of Gonadotropin-Releasing Hormone, Leptin Hormone, Luteinizing Hormone, Follicle-Stimulating Hormone, and Obesity in Polycystic Ovarian Syndrome,” International Journal of Scientific Research in Biological Sciences, Vol.11, Issue.4, pp.1-6, 2024.

Additional comments

The correct way to write "et al." is in italics

Reviewer 4 ·

Basic reporting

The manuscript is in accordance with the aims and scope of the journal, however the language needs to be improved. Some of the literatures are over 10 years old; please update to a maximum of 10 years if possible. There are misspelled words and spacing is inadequate (such as line 75, 81, 89, etc).More literatures need to be added in the introduction and discussion regarding the possible effect of high LH/FSH ratio on oocyte or embryo quality. Also add studies that differ from your research result in the discussion to add more strength in you manuscript.

Experimental design

This study involved quite a number of patients with PCOS undergoing IVF. However, only 95 people underwent mild stimulation. In subgroup analysis, only 23 patients had an LH/FSH ratio >2. This can give rise to sampling bias and potentially impair the accuracy of the analysis. Furthermore, because this research is a retrospective study, subjects were not given random stimulation methods. This could lead to subject selection bias.

Validity of the findings

The findings are well explained, and appropriate statistical methods are used. Conclusions are in accordance with the findings, but limitations that could influence the results and suggestions for improvement are better included. Please explain the mechanism and cite the study regarding progesterone resistance in PCOS (line 196). Please explain more about the possible positive effect of suppression of LH by GnRH agonist on the quality of blastocyst compared to mild stimulation protocol and add studies related to these explanations (basic or clinical studies) (line 208-210)

---

## Round 0.2 · accepted · Accept

After the revisions in response to the reviewers' comments, I would like to inform you that in my opinion your article is now Acceptable. Your manuscript has been accepted for publication in PeerJ.

Reviewer 1 ·

Basic reporting

Please check the general comments listed below.

Experimental design

Please check the general comments listed below.

Validity of the findings

Please check the general comments listed below.

Additional comments

The authors did not sufficiently capitalize on the feedback provided by the reviewers.

The limitations addressed within the discussion of the manuscript were minimal, especially considering the significant deficiencies identified in the clinical management of patients with polycystic ovary syndrome (PCOS) and infertility undergoing controlled ovarian stimulation for in vitro fertilization (IVF) procedures. Numerous published studies have demonstrated that women with PCOS and high ovarian reserve—characterized by elevated levels of anti-Müllerian hormone (AMH) and increased antral follicle count—exhibit a heightened risk of ovarian hyperstimulation syndrome (OHSS). Therefore, these patients should be managed utilizing antagonist protocols combined with agonist triggering and should follow a strategy of cryopreservation for all embryos, followed by subsequent transfer of the thawed embryos. This approach represents the current best practice in the field; however, the center in question continues to transfer fresh embryos and employs agonist protocols with human chorionic gonadotropin (hCG) triggering, which is inconsistent with contemporary medical standards and safety.

Furthermore, the authors did not engage with any of the relevant bibliographic references provided, despite ending the discussion with a non-original idea for a future perspective, discussed by one of the suggested bibliographic references and which was not considered, which could have facilitated a more robust discussion regarding the limitations of the study. Consequently, I find myself compelled to recommend the outright rejection of the manuscript.

·

Basic reporting

1.The language of this manuscript is clear and well-structured, effectively conveying the research findings. The authors have done a commendable job in presenting their work with clarity and precision.
2.In academic writing, to format "et al." in italics within citations, you can simply write it as follows:
Example:
(Dineen et al., 2017).
3. The figures and tables title are appropriate.

Experimental design

1.The retrospective analysis approach is highly suitable for this type of study, allowing for comprehensive insights based on existing data.

Validity of the findings

The results of the current study are both statistically and clinically valid.

Reviewer 4 ·

Basic reporting

The manuscript is in accordance with the aims and scope of the journal, and everything is in line with my queries

Experimental design

Everything is in line with my queries

Validity of the findings

Everything is in line with my queries